# Performance Analysis and Experimental Research of a Dual-Vibrator Traveling Wave Ultrasonic Motor

**DOI:** 10.3390/mi14081610

**Published:** 2023-08-16

**Authors:** Zhaopeng Dong, Liang Xu

**Affiliations:** 1College of Engineering Science and Technology, Shanghai Ocean University, Shanghai 201306, China; 2Suzhou Zhitu Technology Co., Ltd., Shanghai 201106, China; seablueboy@163.com

**Keywords:** ultrasonic motor, dual vibrators, driven mode, theoretical analysis, experimental investigation

## Abstract

In order to facilitate the widespread application of ultrasonic motors, it is essential to conduct a quantitative study aimed at enhancing their performance. The present paper provides a comprehensive theoretical analysis of an ultrasonic motor equipped with dual vibrators, enabling operation in both the single-driven and dual-driven modes, thereby enhancing versatility in terms of performance adjustment. This study provides a detailed examination of the motor’s unique performance characteristics and its varying output responses to different driving signals. Experimental investigations are conducted in both the single-driven and dual-driven modes to validate theoretical predictions. The results demonstrate that the motor exhibits a maximum speed, torque, and power that are 1.59, 1.28, and 1.62 times higher than those of the single-driven stator, respectively. A conclusion can be drawn that the motor will attain the desired performance when operated in the appropriate driven mode.

## 1. Introduction

Ultrasonic motors (USMs) offer a multitude of advantages in comparison to electromagnetic motors, such as the ability to generate motion at the nanometer scale; being compact, energy saving, and large driving; and can hold forces in small sizes and conduct silent nonmagnetic operations, especially at the millimeter scale [1,2,3,4,5]. The aforementioned advantages make USMs highly suitable for ultraprecise positioning in specialized environments and endow them with significant potential in the fields of aerospace mechanisms, optical instruments, biological equipment, and micro-electromechanical systems [6,7]. However, the performance of a general ultrasonic motor is limited to a single characteristic under specific preload conditions, which may not be sufficient to meet the diverse and dynamic application requirements. The continuous utilization of ultrasonic motors inevitably leads to performance degradation. The adjustment of preload is typically required to modify the contact state between the stator and rotor to maintain a stable performance. But in certain instances where the motor is utilized within a closed environment, this drawback may result in the motor’s inability to perform optimally.

The issues of inflexible motor performance and limited applicability have been addressed through numerous studies focusing on innovative and optimized designs for ultrasonic motors. One typical approach involves increasing the number of vibrators, utilizing Langevin transducers or double excitation vibrators. For example, Kurosawa et al. [8] proposed a sandwich-type ultrasonic motor consisting of two vibrators for high-thrust applications. Satonobu et al. [9] introduced a symmetric hybrid transducer-based ultrasonic motor design. Iula and Pappalardo [10] developed a high-power traveling wave ultrasonic motor utilizing multiple transducers. Bai et al. [11] initially proposed the concept of a novel structured motor utilizing two vibrators in which the rotor’s rotational speed is locked by the phase-velocity difference between two traveling waves propagating on both the stator and rotor. Pang [12] designed a plate ultrasonic motor under a dual-mode coupling drive, where the motor is driven by superimposing two frequencies applied to the vibrator. Burhanettin [13] used a transient analysis feature to analyze the behavior of the stator under a dual-source, dual-frequency driving condition. Meanwhile, in terms of drive control, Chen [14] designed a drive circuit that simultaneously employs both the driving frequency and phase modulation control schemes. Tan [15] designed a rotor speed stability control strategy for a two-phase traveling wave ultrasonic motor based on voltage and current double feedback. These studies demonstrate that increasing the number of vibrators and adopting a reasonable driving strategy are effective methods to enhance motor performance.

The present study employed a dual-vibrator ultrasonic motor, enabling the autonomous manipulation of the vibration properties associated with each individual vibrator. The performance of the motor is determined by the interaction between these two vibrators, providing a range of output possibilities depending on the specific combination of stator and rotor vibrations. The initial section presents a comprehensive overview of the operational principle behind this innovative design, followed by a meticulous analysis of the factors that influence the motor’s performance. Subsequently, based on an examination of the key factors affecting the motor’s performance, strategies for adjusting the interaction between the two vibrators are explored. Different output performances of the motor under varying vibration combinations are analyzed. Subsequently, experimental tests are conducted to validate the theoretical results of the analysis.

## 2. Working Principle

The proposed motor features a structure that incorporates two vibrators, namely, the vibrating stator and vibrating rotor. By supplying two-phase sinusoidal voltages with a 90° phase difference to piezoelectric ceramic groups A and B, respectively, a traveling wave is generated. In this motor design, not only does the stator generate one traveling wave but the rotor also generates another. Four-phase sinusoidal voltages, each with specific phase differences, are applied to the piezoelectric ceramic groups A, B, A’, and B’, respectively. Signals A ($\sin\left(\omega t\right)$) and B ($\cos\left(\omega t\right)$) are applied to the stator, while signals A’ ($\sin\left(\omega t+\varphi \right)$) and B’ ($\cos\left(\omega t+\varphi \right)$) are applied to the rotor. The excitation of piezoelectric ceramics results in the generation of two traveling waves within the elastic bodies of both the stator and rotor, which mutually propagate forward upon contact. Refer to Figure 1 for a depiction of the vibration and contact between these two vibrators.

### 2.1. Velocity Model

According to Kirchhoff’s plate theory, the equation of motion for a surface point on the stator can be expressed as follows:(1)$$\frac{u_{z}{}^{2}}{\xi^{2}}+\frac{u_{x}{}^{2}}{{\left(nh_{0}\xi \right)}^{2}}=1$$

The tangential velocity of a surface point on the stator ($v_{s\_t}$) is being referred to:(2)$$v_{s\_t}=\frac{\partial u_{x}}{\partial t}=-nh_{0}\xi \omega_{n}\sin\left(nx-\omega_{n}t\right)$$
where ξ is the vibrating amplitude, n is the number of traveling waves, $h_{0}$ is the half thickness of the stator, and $\omega_{n}$ is the angular frequency of the actuate signals.

Similar to the motion analysis of the surface point in the stator, the tangential velocity of the point in the rotor can also be expressed as follows:(3)$$v_{r\_t}=-nh_{0}{}^{\prime }\xi^{\prime }\omega_{n}{}^{\prime }\sin\left(nx-\omega_{n}{}^{\prime }t-\varphi \right)$$
where $\xi^{\prime }$ is the vibration amplitude of the rotor, $h_{0}{}^{\prime }$ is half the thickness of the rotor, and $\omega_{n}{}^{\prime }$ is the angular frequency of the actuate signals in the rotor.

When the two traveling waves propagate in the same direction, the relative motion directions of the two traveling waves and the particle motion directions on the surfaces of the two oscillators are as shown in Figure 2a. Assuming the rotation of the motor reaches a steady state, the running speed of the motor, $v_{m}$, has the following quantitative relationship with the rotor particle speed, $v_{r\_t}$, and the stator particle speed, $v_{s\_t}$:(4)$$v_{m}-v_{r\_t}=v_{s\_t}$$
(5)$$\begin{array}{l} v_{m}=v_{r\_t}+v_{s\_t}\\ =-kh_{0}{}^{\prime }\xi_{0}{}^{\prime }\omega \cos\left(kx-(\omega t+\varphi )\right)-kh_{0}\xi_{0}\omega \cos\left(kx-\omega t\right)\\ =-\left(kh_{0}{}^{\prime }\xi_{0}{}^{\prime }\omega \cos\left(kx-(\omega t+\varphi )\right)+kh_{0}\xi_{0}\omega \cos\left(kx-\omega t\right)\right) \end{array}$$

In the formula, ‘−’ means that the motor’s running direction is opposite to the traveling wave’s direction.

Under this movement direction of the two traveling waves, the motor can obtain the “combined speed”.

The no-load speed of the motor is:(6)$$v_{m\_\max}=-k\omega (h_{0}{}^{\prime }\xi_{0}{}^{\prime }+h_{0}\xi_{0})$$

In contrast, when the two traveling waves propagate in the opposite directions, the relative motion directions of the waves and the particle motion directions on the surfaces of the oscillators are depicted in Figure 2b. Assuming the rotation of the motor reaches a steady state, the running speed of the motor, $v_{m}$, has the following quantitative relationship with the rotor particle speed, $v_{r\_t}$, and the stator particle speed, $v_{s\_t}$:(7)$$v_{m}+v_{r\_t}=v_{s\_t}$$
(8)$$\begin{array}{l} v_{m}=v_{s\_t}-v_{r\_t}\\ =-kh_{0}\xi_{0}\omega \cos\left(kx-\omega t\right)+kh_{0}{}^{\prime }\xi_{0}{}^{\prime }\omega \cos\left(kx-(\omega t+\varphi )\right)\\ =-\left(kh_{0}\xi_{0}\omega \cos\left(kx-\omega t\right)-kh_{0}{}^{\prime }\xi_{0}{}^{\prime }\omega \cos\left(kx-(\omega t+\varphi )\right)\right) \end{array}$$

Under this movement direction of the two traveling waves, the motor can obtain the “differential speed”.

Based on the aforementioned analysis, the dual-vibrator ultrasonic motor exhibits distinct motion characteristics when the traveling waves of its two vibrators move in different directions. This is a significant feature of this type of motor: when both traveling waves move in the same direction, the motor speed equals the sum of their speeds; conversely, when they travel in opposite directions, the motor speed is determined by their difference.

### 2.2. Contact Model

The cylindrical model is used to analyze the contact surface of the two modes as follows:

Referring to the analysis method of Hertzian contact theory, the contact area (−L/2,L/2) is divided into the driving area (−D/2,D/2) and the hindering area (−L/2,D/2) (D/2,L/2). The contact analysises inclued force analysis and velocity analysis are shown in Figure 3a,b.

From the mechanics, the contact force is defined as:(9)$$F_{N}=k\omega \int_{-L/2}^{L/2}p\left(x\right)dx$$
where k is the number of traveling waves, ω is the traveling wave angular frequency, and p(x) is the stress distribution function.

The stress distribution function can be expressed as:(10)$$p\left(x\right)=k\frac{\pi F_{N}}{n\omega \lambda \left[\sin\left(\frac{\pi L}{\lambda }\right)-\frac{\pi L}{\lambda }\cos\left(\frac{\pi L}{\lambda }\right)\right]}\left[\cos\left(\frac{2\pi x}{\lambda }\right)-\cos\left(\frac{\pi L}{\lambda }\right)\right]$$

The radial pressure ($F_{Z}$) and the axial driving force ($F_{T}$) generated in the contact layer can be expressed as:(11)$$F_{Z}=k\omega \int_{-L/2}^{L/2}p\left(x\right)\cos\left(\frac{2\pi x}{\lambda }\right)dx$$
(12)$$F_{T}=\mu k\omega \int_{-L/2}^{L/2}p\left(x\right)\mathrm{sgn}\left[v_{s\_t}\left(x\right)-v_{r}\left(x\right)\right]\cos\left(\frac{2\pi x}{\lambda }\right)dx$$
where μ is the coefficient of friction, and $\mathrm{sgn}[v_{s\_t}(x)-v_{r}(x)]=\left\{\begin{array}{cc} 1, & v_{s\_t}(x)>v_{r}(x)\\ 0, & v_{s\_t}(x)=v_{r}(x)\\ -1, & v_{s\_t}(x)<v_{r}(x) \end{array}\right.$.

Among them, $v_{r}\left(x\right)$ is a process quantity, and its physical meaning is a constant velocity adhesion point. In the motor, it is the result of the combined effect of the motor speed and the rotor particle moving speed, as follows:(13)$$v_{r}\left(x\right)=\left\{\begin{array}{c} v_{m}-v_{r\_t}\left(x\right)\begin{array}{cc} & Condition\begin{array}{cc} A & \end{array} \end{array}\\ v_{m}+v_{r\_t}\left(x\right)\begin{array}{cc} & Condition\begin{array}{cc} B & \end{array} \end{array} \end{array}\right.$$
(14)$$v_{m}=\frac{2\pi R_{0}N}{60}$$
where N is the motor speed (rpm), which is the rotor contact radius; Condition A means that the traveling waves of the two oscillators travel in the same direction; and Condition B means that the traveling waves of the two oscillators travel in different directions.

The output torque is:(15)$$T=F_{T}R_{0}=\mu k\omega_{d}R_{0}\int_{-L/2}^{L/2}p\left(x\right)\mathrm{sgn}\left[v_{s\_t}\left(x\right)-v_{r}\left(x\right)\right]\cos\left(\frac{2\pi x}{\lambda }\right)dx$$

From the above analysis, it can be seen that the motor drive area is divided by the constant velocity adhesion point, which is divided into two parts: drive area and obstruction area. The motor output torque is the result of the synthesis of the driving effect from the drive area and the obstruction effect of the obstruction area. The position where the motor’s limit torque appears is where the constant velocity adhesion point falls on the edge of the contact area. At this time, the speed of all stator particles in the contact area is higher than or equal to the rotor movement speed. Therefore, the limit torque at this time can be expressed as:(16)$$T_{\max}=\mu k\omega_{d}R_{0}\int_{-L/2}^{L/2}p\left(x\right)\cos\left(\frac{2\pi x}{\lambda }\right)dx$$

The efficiency of the Hertzian contact model can be expressed by the ratio of the output power of the motor to the input power:(17)$$\eta_{m}=\frac{P_{m}}{P_{i}}=\frac{P_{m}}{P_{m}+P_{1}}$$
where $P_{1}$ is the energy loss generated when the stator and rotor are in contact with each other.
(18)$$P_{1}=T\left(\omega_{s}-\omega_{r}\right)=2\mu R_{0}\int_{0}^{x_{0}}\mathrm{sgn}\left[\Delta \omega \left(x\right)\right]p\left(x\right)\Delta \omega \left(x\right)dx$$

In the formula, $\omega_{s}$ and $\omega_{r}$ are the angular velocities of the stator and rotor, and $\mathrm{sgn}[\Delta \omega \left(x\right)]$ is the sign function:(19)$$\mathrm{sgn}[\Delta \omega \left(x\right)]=\left\{\begin{array}{cc} 1, & \omega_{s}\left(x\right)>\omega_{r}\\ 0, & \omega_{s}\left(x\right)=\omega_{r}\\ -1, & \omega_{s}\left(x\right)<\omega_{r} \end{array}\right.$$

Expand the abovementioned piecewise integral:(20)$$\begin{array}{l} P_{1}=2k_{r}A_{p}\mu R_{0}\int_{0}^{x_{r}}\left[\cos\left(kx\right)-\cos(kx_{0})\right]\left[\omega_{s\max}\cos\left(kx\right)-\omega_{r}\right]dx\\ \begin{array}{cc} & - \end{array}2k_{r}A_{p}\mu R_{0}\int_{x_{r}}^{x_{0}}\left[\cos\left(kx\right)-\cos(kx_{0})\right]\left[\omega_{s\max}\cos\left(kx\right)-\omega_{r}\right]dx \end{array}$$

Introducing the function:

$\psi (x)=\frac{1}{2}kx\omega_{s\max}+\frac{1}{4}\omega_{s\max}\sin(2kx)-\left[\omega_{s\max}\cos\left(kx_{0}\right)+\omega_{r}\right]\sin(kx)-kx\omega_{r}\cos\left(kx_{0}\right)$ and $\phi (x)=\sin(kx)-kx\cos(kx_{0})$.

Simplify the above formula to obtain:(21)$$P_{1}=\frac{\mu F_{N}R_{0}}{\phi (x)}[2\psi (x_{r})-\psi (x_{0})]$$

Therefore, the efficiency of the motor is:(22)$$\eta_{m}=\frac{T\omega_{r}}{\frac{\mu F_{N}R_{0}}{\phi (x_{0})}\left[2\psi (x_{r})-\psi (x_{0})\right]+T\omega_{r}}$$

### 2.3. Advantages in Performance

(1)Speed advantage

Based on the design of the dual-vibrator and double traveling waves, the motor is capable of achieving not only more flexible speed adjustment but also different speed performances such as “combined speed” and “differential speed”. The range of speed adjustment is wider than that of traditional motors, as shown in Figure 4.

(2)Torque advantage

In terms of driving force, the difference between dual-vibrator ultrasonic motors and traditional motors is compared. Assuming a constant output speed requirement, the motor speeds shown in Figure 5a,b are equal. The relationship among the contact point velocities shown in Figure 5a,b can be expressed as follows:$$v_{s\_t(b)}<v_{s\_t(a)}$$

According to velocity ellipse distribution theory, the position of the contact point in Figure 5b will be lower than that in Figure 5a, which will inevitably lead to the driving area in the contact area of Figure 5b being larger than that in Figure 5a which, in turn, leads to a larger driving force, as shown in Figure 6. Under the requirement of the same motor speed, the dual-vibrator ultrasonic motor can output a larger driving torque than traditional motors.

## 3. Experiment Analysis

### 3.1. Motor Drive Strategy

The drive strategy design of the dual-vibrator traveling wave ultrasonic motor is mainly composed of the control part and the drive part in the hardware. The overall system scheme is shown in Figure 7. The control part includes a PC and an FPGA controller, and the driving part includes a driver board, a matching board, and a detection module. LabVIEW2020 was used on the PC to carry out serial communication (RS232), complete functions of human–computer interactions, and monitor the operating status of the ultrasonic motor. The PWM wave output by the FPGA controller can realize the related control logic and analyze and process the feedback signal. The drive board included a bridge drive circuit, a full-bridge inverter circuit, an optocoupler isolation circuit, a current detection circuit, and a protection circuit. The matching board was a set of boost matching circuits, which converts the PWM wave carrying control information output by the driver board into a strong current signal and then generates two high-frequency sinusoidal voltages with a certain phase difference through the boost matching circuit to drive the ultrasound vibrator. The detection module was a set of current detection circuits, which were integrated on the drive board to facilitate the layout. The current information on the inverter bridge was detected using a current sensor, and it was sampled with A/D and transmitted to the FPGA controller using I2C communication.

The experiment platform is shown in Figure 8, which included the host computer PC, power board, matching board, AD sampling, oscilloscope, and regulated power supply. The drive control parameters of the motor are being debugged in the picture. The oscilloscope shows the gate voltage of the four MOSFET switches of one of the full bridges when the motor is working. In order to facilitate the control of the start and stop of the motor, a hard switch is added to individually control the 12 V power supply voltage of the inverter bridge. Under the premise of not affecting the power supply of the driving part, the power supply of the motor can be directly disconnected. At the same time, the soft start–stop button in the LabVIEW program interface of the host computer can also realize the start–stop control of the motor. It uses PWM to control the start and stop of the motor. Therefore, during the experiment, the best order to drive the motor is to turn on the software first to determine whether the PWM wave output on the power board is normal and then close the hard switch 1 and switch 2 to take into account the safety and stability of the motor startup. A laser velocimeter is used to measure the motor speed. The dynamometer is used to measure the output torque.

### 3.2. Motor Performance Test

(1)Single-driven experiment (using the stator)

The ultrasonic motor operates on the same principle as a conventional traveling wave motor, with a vibrating stator driving the rotation of the rotor. Impedance analysis experiments preliminarily determined that the stator’s drive frequency is 47.5 kHz. The experimental analysis, as shown in Figure 9, clearly illustrated the relationship between the drive frequency and motor speed.

According to the relationship diagram between driving frequency and output speed, it can be observed that the optimal driving frequency for the stator is approximately 47.5 kHz, which aligns with both the designed modal frequency and impedance analysis results. When the ultrasonic motor stator is near the optimal operating frequency, the motor speed is higher, and when it deviates from the optimal operating frequency, the motor speed decreases significantly, which conforms to the theory.

After determining the optimal drive frequency of the stator, when the preload and the drive voltage are fixed at $F_{N}=150\;N$ and $V_{p-p}=300\;V$, the motor stator is driven separately, and the output performance of the motor is obtained, as shown in Figure 10.

It can be seen from Figure 10 that when only the stator is driven, the motor can obtain a maximum drive torque of T=0.65 N⋅m, and the maximum output speed is v=30.2 rpm.

(2)Single-driven experiment (by rotor)

Different from the traditional ultrasonic motor working mode, the rotor of the proposed motor can be driven separately to generate ultrasonic vibration to act on the stator, which is fixed to the base of the motor. Because of the existence of force and reaction force, the rotor is in forced rotation. The experimental operation process is similar to the one driving the stator alone. The suitable operating frequency for the rotor is tested first, and the motor’s performance is tested and analyzed at this operating frequency.

Figure 11 shows the relationship between the drive frequency and motor speed when the rotor is driven separately:

Figure 11 shows the relationship between the drive frequency and output speed. The optimal driving frequency for the motor to achieve maximum speed is approximately 48 kHz, with the single-driven rotor. The modal frequency deviates by approximately 0.5 kHz from the designed value, which may be attributed to several factors, including, firstly, potential back-buckling of the rotor on the stator, where mechanical connection strength and system rigidity could marginally impact its operational frequency; secondly, the utilization of conductive brushes for electrical signal application on the rotor, which might influence the electrical signal transmission. Because the frequency difference is not large and the designed working principle requires the same driving frequency for the stator and rotor, it can still be driven with a frequency of 47.5 kHz (the same as the stator).

Figure 12 shows a performance analysis diagram of the motor with a single-driven rotor. It can be seen that when the proposed motor is only driven by the rotor, the motor can obtain a maximum drive torque of T=0.5 N⋅m. The maximum output speed is v=20.1 rpm, which is not the same as the output performance when the motor is only driven by the stator. The discrepancy primarily arises from the vibration characteristics and the installation methods of both the stator and rotor.

(3)Dual-driven experiment

Based on the working mechanism of the dual-vibrator motor, if the proposed motor with dual traveling waves is wanted to operate stably, the double vibrators should be applied with a similar driving frequency. From an analysis of the above experimental results, it can be determined that the stator and rotor can behave well at a working frequency of 47.5 kHz and show better vibration characteristics. So this was the driving frequency used for the stator and rotor in the dual-driven working mode.

The experimental results demonstrate that the motor exhibits superior performance in the dual-driven working mode when both traveling waves propagate in the same direction, compared to when the stator and rotor are driven separately. The output is the summation of the two single-driven modes, which aligns with the conclusion of theoretical analysis. To clearly demonstrate the numerical performance of this synthetic effect, two comparisons are presented to evaluate the motor’s performance (i.e., load characteristics) in both the single-driven and dual-driven modes. In addition, because of the limitations of the driving mode, the driving ability of the rotor is relatively weak, the load torque range that it can drive is 0<T<0.5 N⋅m. Therefore, the performance comparison analysis only considered the load torque under the situation 0<T<0.5 N⋅m, as shown in Table 1.

The analysis in Figure 13 and Table 1 reveals that, under the same torque requirement, the motor’s performance (motor speed) can approximately achieve the combined performance of a single-driven stator and a single-driven rotor. The data in Table 2 demonstrate a significant enhancement in both the maximum speed and maximum torque achieved by the motor in a dual-driven configuration compared to a single-driven one. Specifically, the maximum speed increased by 59%, while the maximum torque experienced an increment of 28%. The dual-vibrator ultrasonic motor significantly enhances the motor’s output performance (compared to the single-driven mode) and expands its range of performance adjustment.

Comparing the output power of the motor in the dual driven and single driven modes by stator, as shown in Figure 14:

The maximum power of the motor in the dual-driven mode was 1.52 W, as observed from Figure 14, while it was 0.94 W in the single-drive in the stator mode. The power in the dual-driven mode was 1.62 times higher than that in the single-drive in the stator mode, thus confirming the effectiveness of the dual-vibrator mechanism.

From the above method of judging the “same direction” of the double traveling waves, it can be understood that if the stator and rotor are driven separately to make the motor rotate in the same direction, this means that the stator traveling wave and the rotor traveling wave are in the “opposite direction” at this time. Under this condition, a motor performance comparison between the dual-driven working mode and the two single-driven working modes was also conducted, as shown in Figure 15.

Similar to the above analysis, in order to clearly analyze the numerical performance of this “difference motion”, we made a comparison between the numerical difference of the motor performance under two single-driven modes and the performance of the motor under the dual-driven modes. The output speed comparison is shown in Table 3.

The experimental results demonstrate that when the two traveling waves propagate in the opposite direction, the performance of the motor in the dual-driven working mode was worse than that of the single-driven by the stator or the single-driven by the rotor, which is represented by the difference effect of the two. It can be found from Table 3 that when the difference motion is formed, the motor speed is large and the torque is smaller, and the dual-driven can also show a relatively stable performance. At this time, there is a large speed fluctuation when the motor is dual driven. Therefore, this working mode (two traveling waves propagating in the opposite direction) of the dual-vibrator ultrasonic motor is only a special speed adjustment method, which is rarely used in regular applications.

## 4. Conclusions

The aforementioned experimental analysis leads to the following conclusions:(1)The dual-vibrator ultrasonic motor features two operational modes: single-driven and dual-driven. In the former, either the stator or the rotor can be driven independently to generate power or motion, respectively; in the latter, both components are simultaneously activated to produce two traveling waves propagating in either parallel or antiparallel directions. The proposed motor demonstrates diverse output performances in response to different driving conditions, as necessitated.(2)In the dual-driven working mode, when the two traveling waves are in “resultant” motion, the motor’s performance can achieve approximately 90% of the combined output of the single-driven mode by stator and the single-driven mode by rotor; whereas, when the two traveling waves are in “differential” motion, a more pronounced differential effect is observed.(3)The maximum speed, torque, and power of the dual-driven mode motor are, respectively, 1.59, 1.28, and 1.62 times that of the single-driven mode by stator.

The effectiveness of the dual-vibrator motor in improving the performance was confirmed by performance experiments. However, there is still significant room for improvement in its overall performance, including:(1)Because of concerns regarding the impact of friction materials on the interaction between the two traveling wave modes, no friction layer material was incorporated onto the contact surface of the two vibrators, which adversely affects the motor’s performance.(2)The efficiency of energy conversion is directly correlated with the quality of the piezoelectric ceramic bonding process. Any imperfection in this process can significantly impact the motor’s performance.(3)The subsequent improvement of the brush structure should take into account the contact force application mechanism between the stator and rotor, as it is crucial for ensuring the motor’s stable performance.

## Figures and Tables

**Figure 1 micromachines-14-01610-f001:**
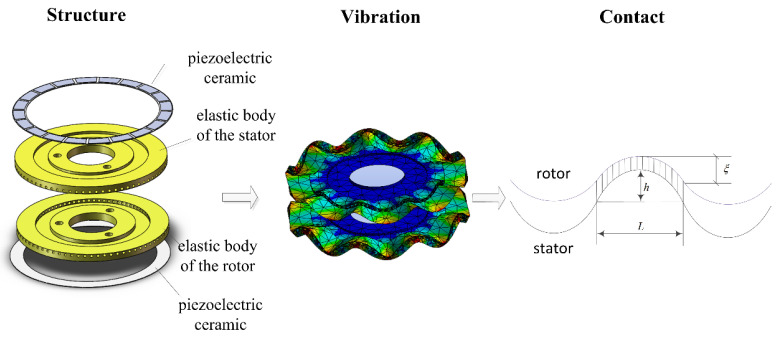
Vibration and contact of the two vibrators.

**Figure 2 micromachines-14-01610-f002:**
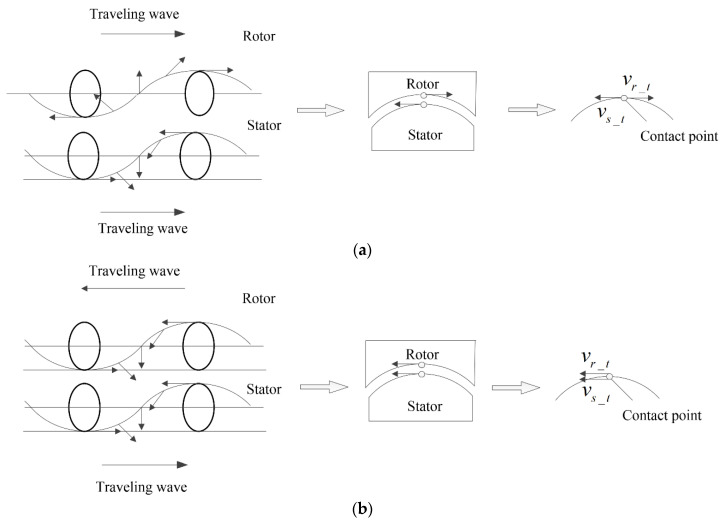
Particle velocity synthesis. (**a**) Two traveling waves propagate in the same direction. (**b**) Two traveling waves propagate in the opposite directions.

**Figure 3 micromachines-14-01610-f003:**
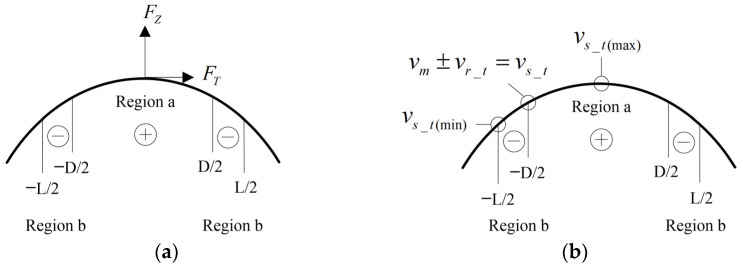
Analysis of the contact surface. (**a**) Force analysis. (**b**) Velocity analysis.

**Figure 4 micromachines-14-01610-f004:**
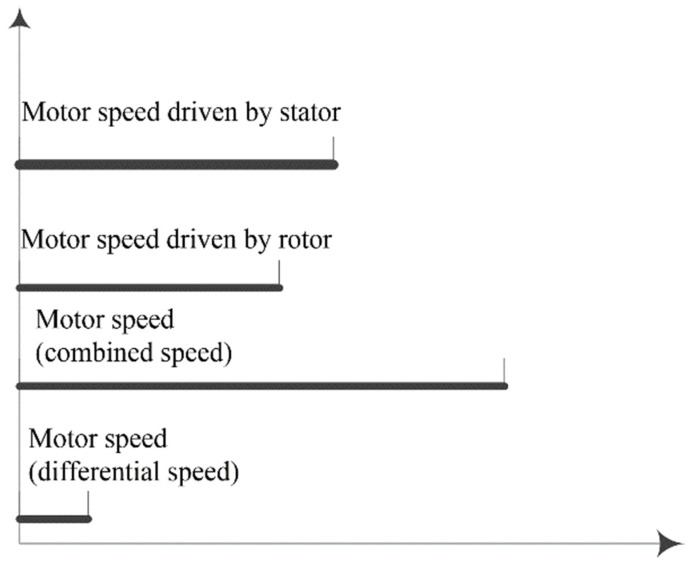
Adjustable range of speed in the dual-vibrator motor.

**Figure 5 micromachines-14-01610-f005:**
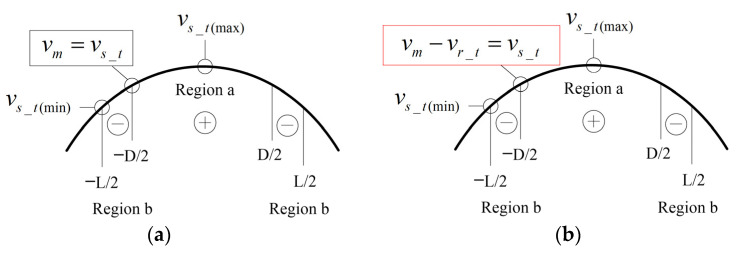
Comparison of the contact point speeds between the traditional motor and dual-vibrator motor. (**a**) Particle velocity in the traditional motor. (**b**) Particle velocity in the dual-vibrator motor.

**Figure 6 micromachines-14-01610-f006:**
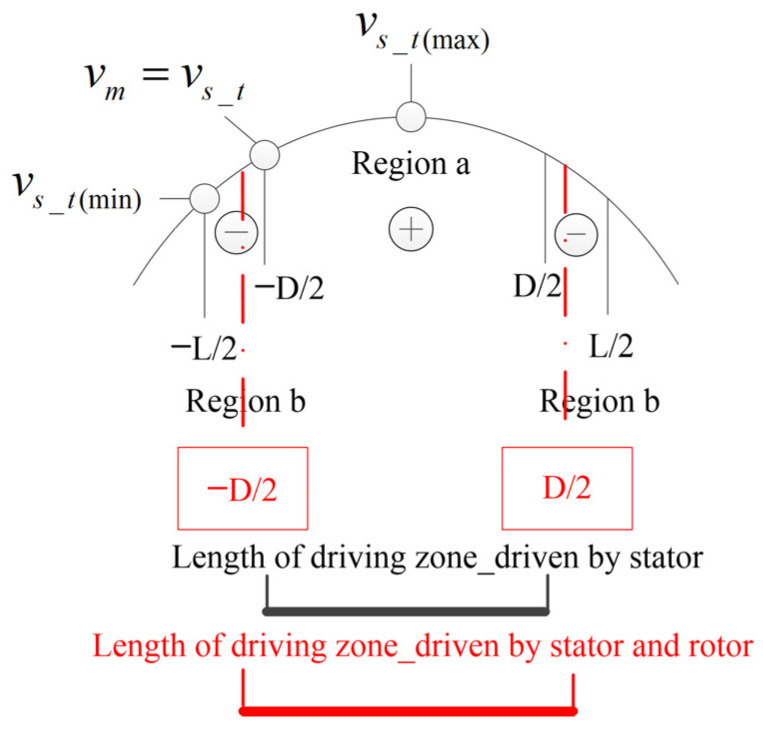
Comparison of the drive zone length between the traditional motor and the dual-vibrator motor.

**Figure 7 micromachines-14-01610-f007:**
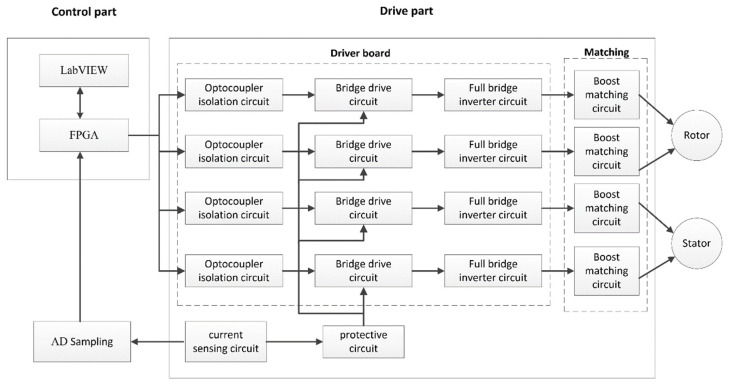
General block diagram of the ultrasonic motor drive power system.

**Figure 8 micromachines-14-01610-f008:**
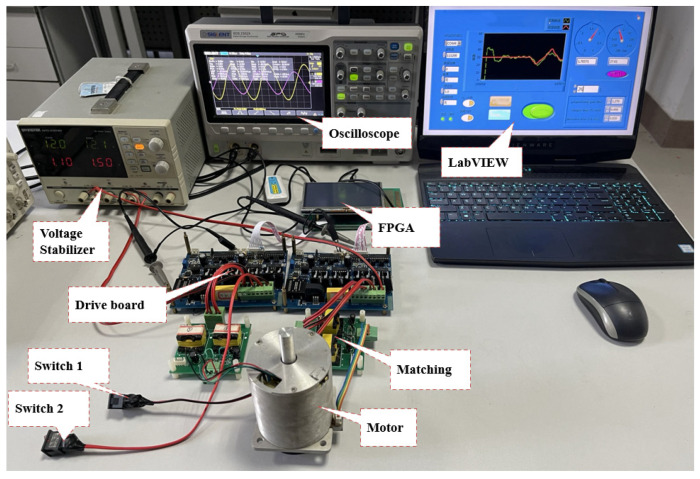
Experimental platform of the motor driving.

**Figure 9 micromachines-14-01610-f009:**
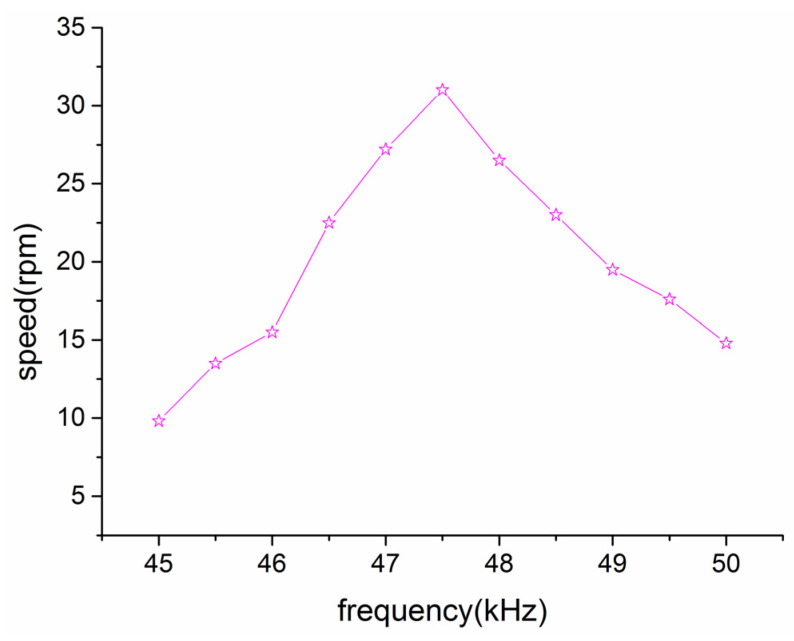
Optimal drive frequency of the stator.

**Figure 10 micromachines-14-01610-f010:**
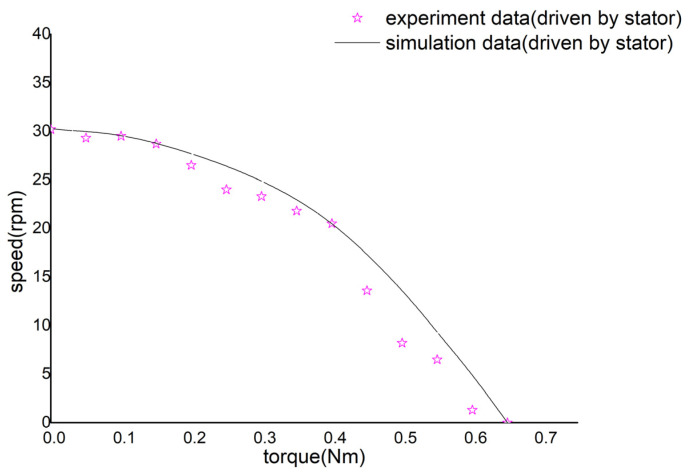
Motor’s performance with the single-driven stator.

**Figure 11 micromachines-14-01610-f011:**
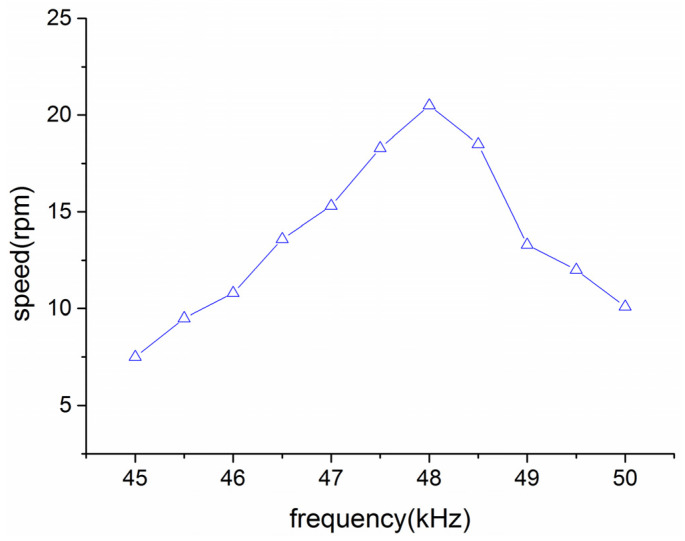
Optimal drive frequency of the rotor.

**Figure 12 micromachines-14-01610-f012:**
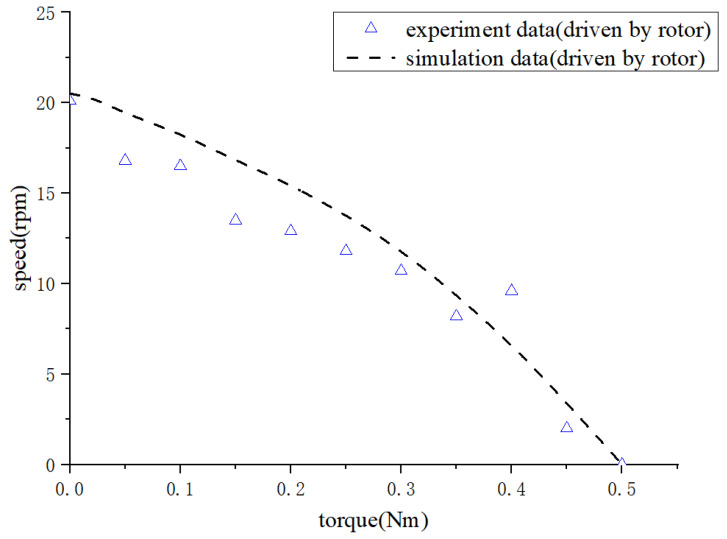
Motor’s performance with the single-driven rotor.

**Figure 13 micromachines-14-01610-f013:**
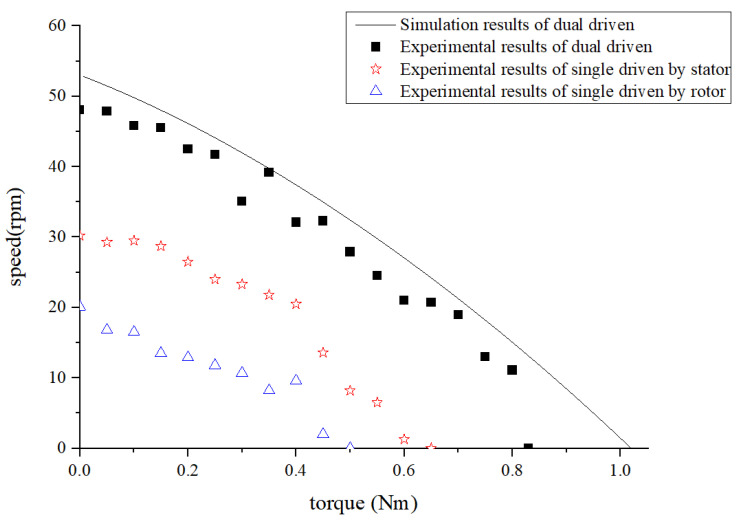
Output performance of the motor in the double traveling wave’s “same direction”.

**Figure 14 micromachines-14-01610-f014:**
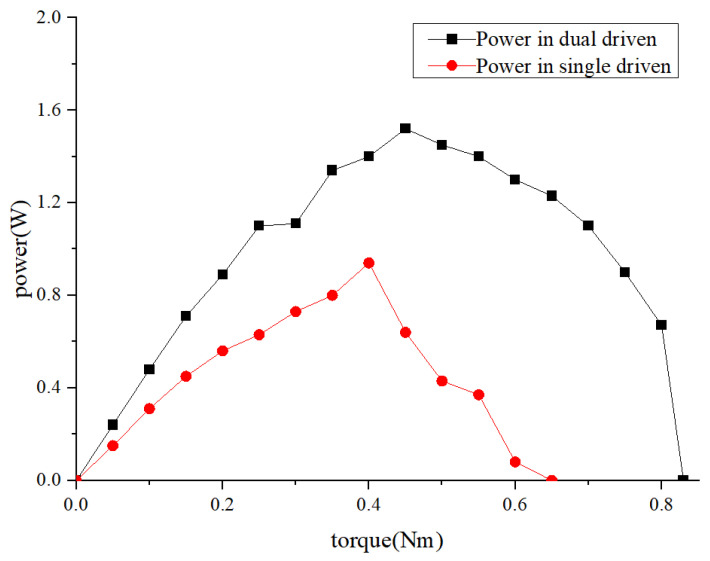
Motor power comparison between the dual-drive mode and single-driven mode.

**Figure 15 micromachines-14-01610-f015:**
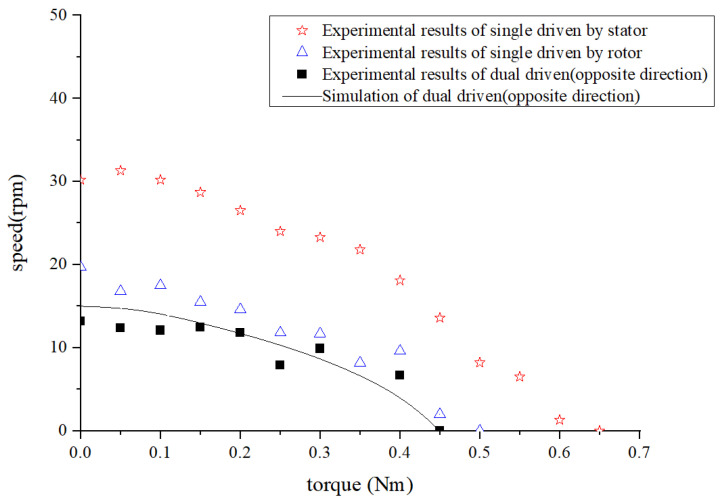
Output performance of the motor for the two traveling wave in the opposite direction.

**Table 1 micromachines-14-01610-t001:** Comparison of the motor’s performance in the single-drive and dual-drive modes under the same torque (two traveling wave in the same direction).

	Nm	Single Driven by Stator (A)	Single Driven by Rotor (B)	Dual Driven	Sum of A and B	Relative Difference
rpm						
T=0		30.2	20.1	48.1	50.3	4.37%
T=0.1		29.5	16.5	45.8	46	0.43%
T=0.2		26.5	12.9	42.5	39.4	7.87%
T=0.3		24.8	10.7	35.1	35.5	1.13%
T=0.4		22.4	9.6	32.1	32	0.31%

**Table 2 micromachines-14-01610-t002:** Comparison of the motor’s performance in the single-driven and dual-driven modes.

	Single Driven by Stator	Dual Driven	Increment
Maximum speed v (rpm)	30.2	48.1	59%
Maximum torque T (Nm)	0.65	0.83	28%

**Table 3 micromachines-14-01610-t003:** Comparison of the motor’s performance in the single-driven and dual-driven modes under the same torque(two traveling wave in the opposite direction).

	Nm	Driven by Stator	Driven by Rotor	Dual Driven	Absolute Difference	Lift Amount
rpm						
T=0		30.2	19.7	10.8	10.5	2.78%
T=0.1		29.5	17.5	12.5	12	4%
T=0.2		26.5	13.9	11.8	12.6	6.78%
T=0.3		23.3	11.7	9.9	11.6	17.17%
T=0.4		18.1	9.6	6.7	8.5	21.17%

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
