# Peer review of "Performance Analysis and Experimental Research of a Dual-Vibrator Traveling Wave Ultrasonic Motor"

_micromachines, 2023, doi:10.3390/mi14081610_

Round 1

Reviewer 1 Report

Dear Authors,

The manuscript needs revision before it is published. My comments are listed in the attachment. 

Kind Regards

English should be polished. 

Reviewer 2 Report

Your paper is complete, the performance analysis is interesting and has been validated by measurements.

Please correct Page 6 first sentence R0 is missing

Fig8 there is a shift in the indications (bottom ones)

Section 3.2 Fig 6-11 indicated in the text is not the good one

Page 11 Fig 12 wich frequency has been used? Indicate

correct the the in the sentence "The difference..."

Correct last sentence "as shown in Table 1 Shown"

Page 14 Table 6-5 it is not the good number

Reviewer 3 Report

In this work, a double vibrator ultrasonic motor was proposed and its performance was evaluated via experiments. The topic was interesting and the paper was well organized. Followings are comments for this paper. 

1. In Figure 8, there are items that do not fit well with the photo and description. It is also necessary to use higher resolution photos.

2. It is necessary to provide titles for (a) and (b) in Figures 2 and 5.

3. It is necessary to add descriptive titles using (a) and (b) to the two figures in Figure 3.

4. The introduction part needs reinforcement. It is necessary to add more research papers on related fields and to mention specific analysis results on this in the introduction. This is expected to further emphasize the originality of this paper.

Round 2

Reviewer 1 Report

Dear Authors,

You have modified the manuscript according to my remarks. There is only one small mistake for the reference 13. Please correct it. It should be written below;

[13] B. Koc, L. Von Deyn and B. Delibas, "Dual Source Dual Frequency Drive and Modeling of Resonance type Piezoelectric Motors," ACTUATOR; International Conference and Exhibition on New Actuator Systems and Applications 2021, Online, 2021, pp. 1-4.

Kind Regards

Quality of english is fine